# Clinical Characteristics and Treatment Outcomes of Definitive versus Standard Anti-Tuberculosis Therapy in Patients with Tuberculous Lymphadenitis

**DOI:** 10.3390/jcm8060813

**Published:** 2019-06-07

**Authors:** Yousang Ko, Changwhan Kim, Yong Bum Park, Eun-Kyung Mo, Jin-Wook Moon, Sunghoon Park, Yun Su Sim, Ji Young Hong, Moon Seong Baek

**Affiliations:** 1Division of Pulmonary, Allergy and Critical Care Medicine, Department of Internal Medicine, Kangdong Sacred Heart Hospital, Hallym University College of Medicine, Seoul 05355, Korea; bfspark2@gmail.com (Y.B.P.); ekmopark@gmail.com (E.-K.M.); cozybar@hanmail.net (J.-W.M.); 2Department of Internal Medicine, Jeju National University Hospital, Jeju 63241, Korea; masque70@dreamwiz.com; 3Division of Pulmonary, Allergy and Critical Care Medicine, Hallym University Sacred Heart Hospital, Anyang 14068, Korea; f2000tj@hallym.or.kr; 4Division of Pulmonary, Allergy and Critical Care Medicine, Kangnam Sacred Heart Hospital, Seoul 07441, Korea; sysliver@naver.com; 5Division of Pulmonary, Allergy and Critical Care Medicine, Chuncheon Sacred Heart Hospital, Chuncheon 24253, Korea; mdhong@hallym.or.kr; 6Division of Pulmonary, Allergy and Critical Care Medicine, Dongtan Sacred Heart Hospital, Hwaseong 18450, Korea; wido21@hallym.or.kr

**Keywords:** *Mycobacterium tuberculosis*, tuberculous lymphadenitis, drug susceptibility test, treatment outcome, recurrence

## Abstract

Although it is necessary to culture *Mycobacterium tuberculosis* from tuberculous lymphadenitis (TBL) patients for definitive therapy, based on the drug-sensitivity test (DST), substantial cases remain culture-negative. Limited data are available regarding the treatment outcomes after standard anti-tuberculosis therapy in culture-negative TBL. The aim of this study was to compare the recurrence rates between definitive anti-tuberculosis therapy, based on DST and standard anti-tuberculosis therapy in culture-negative TBL. A multicenter retrospective cohort study was performed from 2011 to 2015 in South Korea. The study population was divided into two groups according to treatment type. A total of 234 patients with TBL were analyzed, who were treated with definitive (84 patients) and standard anti-tuberculosis (150 patients) therapy, respectively. During a 28.0 (24.0–43.0) month follow-up period, nine cases (3.8%) had recurrence of TB after treatment completion. The recurrence rate was not significantly different between the two groups (2/84, 2.4% in definitive anti-tuberculosis therapy group versus 7/150, 4.7% in standard anti-tuberculosis therapy group, *p* = 0.526). The recurrence in all nine cases was diagnosed as clinical recurrence rather than microbiological recurrence. Therefore, culture-negative TBL can be treated with standard anti-TB medication, although DST is not available but clinically stable after initiation of treatment.

## 1. Introduction

Tuberculous lymphadenitis (TBL) is a common form of extra-pulmonary tuberculosis (EPTB), and the leading cause of lymph node enlargement in locations with a high prevalence of tuberculosis [1]. The global incidence of EPTB was up to 14% of the 6.4 million new cases in 2017 [2]. Among these cases, TBL forms a large portion in both developed and developing countries [1,3]. This new–old disease, particularly when it affects the cervical region called as scrofula, has afflicted mankind for thousands of years.

Involvement of lymph nodes by other infectious diseases such as non-tuberculous mycobacteria and non-infectious disease such as malignancy can mimic TBL [1,4]. Thus, it is important to differentiate TBL from other diseases, and this can be achieved by culture of *Mycobacterium tuberculosis* (MTB) or by TB polymerase chain reaction (PCR) using a specimen from an affected lymph node. Isolation of MTB remains the gold standard for a confirmative diagnosis of TBL [1,3]. In addition, the drug-sensitivity test (DST) can be performed concurrently with a culture of MTB in order to provide vital clues to the clinician for correct diagnosis and choice of optimal antimicrobial agents. However, it is more difficult to successfully grow MTB species from TBL specimens than in samples from PTB because of the paucity of bacilli in the former. The rate of positive culture of MTB in a specimen from an affected lymph node is known to range widely from 18% to 62% in patients with TBL according to previous studies [1,5,6]. Although, recent advances in technology such as liquid media culture have led to an improvement in diagnostic yield, substantial numbers of TBL patients still remain culture-negative.

In TBL, the low rate of cultivation of MTB enforces the clinician to treat the disease without information about the DST of the offending organism. This lack of evidence to base drug choice on results in three salient problems. Firstly, there are a similarly significant proportion of patients with drug resistant (DR) MTB bacilli in EPTB as there are in PTB [7]. Secondly, TBL is usually slow to respond to anti-TB treatment and the patient’s condition can paradoxically deteriorate during therapy [8]. Thirdly, in EPTB, there are no documented standard guidelines that stipulate the optimal duration of treatment or outcome like those of PTB [9,10]. Therefore, although it is recommended that 6 months of standard anti-TB treatment is adequate for most cases of TBL, many clinicians have a concern and fear of recurrence of TB in real-life clinical practice, especially in cases treated without DST information [1,3,9,11].

Therefore, we evaluated the treatment outcome between patients with TBL who underwent empirical standard therapy because of the absence of DST results and patients who underwent definitive therapy based on DST results. The aims of this study were to (1) compare the recurrence rate of TBL in the two groups, namely, the standard therapy group versus the definitive therapy group, and (2) to evaluate characteristics of cases with recurrence of TB.

## 2. Materials and Methods

### 2.1. Study Population

We performed a retrospective review of patients diagnosed with TBL in 5 hospitals of Hallym University Medical Centers with >600 beds in the Republic of Korea, an area of intermediate TB burden with a prevalence rate of 143/100,000 persons in 2013 [12]. Medical records between January 2012 and December 2015 were obtained. During the study period, all consecutive patients aged >18 years who had been diagnosed with TBL were screened.

Patients were excluded if they had previously had TB infection that was treated, were transferred to another hospital or lost to follow-up during anti-TB treatment, showed poor compliance with anti-TB medication or had incomplete medical records. Patients were also excluded if they had serological evidence of human immunodeficiency virus (HIV) infection or had been infected with multi-DR TB.

The protocol for this study was approved by the Institutional Review Board of each participating hospital (Kangdong Sacred Heart Hospital, Hallym University Sacred Heart Hospital, Kangnam Sacred Heart Hospital, Chuncheon Sacred Heart Hospital, and Dongtan Sacred Heart Hospital) for the review and publishing of information from the patient records. Informed consent was waived because of the retrospective nature of the study.

### 2.2. Diagnosis of Tuberculous Lymphadenitis

Microbiological and pathological results of affected lymph nodes were used for the diagnosis of TBL. All the lymph node specimens by aspirate or tissue sampling for the Acid-Fast Bacilli (AFB) smears and cultures were processed and pretreated as recommended [13]. The AFB smears were prepared by auramine–rhodamine fluorescence staining and confirmed by Ziehl–Neelsen staining. The MTB cultures were simultaneously performed with solid media, 3% Ogawa media (Eiken Chemical, Tokyo, Japan), and liquid media in the mycobacteria growth indicator tube 960 system (BD Biosciences, Franklin Lakes, NJ, USA). The TB-PCR was conducted using either the Xpert MTB/RIF assay (Cepheid Inc., Sunnyvale, CA) or AdvanSure TB/NTM RT-PCR kit (LG Life Sciences, Seoul, Korea) [14,15]. Interferon-gamma release assay (IGRA) was performed using QuantiFERON-TB Gold in tube test (Cellestis Limited, Carnegie, Victoria, Australia).

The histologic results of the lymph nodes were classified into five groups, by grade of cytomorphological changes, from the highest to lowest grade, based on previously published criteria: grade I, epithelioid granuloma reaction with caseation; grade II, epithelioid granulomatous reaction without caseation; grade III, non-granulomatous reaction with necrosis; grade IV, nonspecific; grade V, inadequate sample [16].

Patients were defined as TBL cases if they met one of the following criteria:positive isolation of MTB in lymph node specimen.positive AFB staining with positive TB-PCR.histological findings compatible with TB (grade I, II, and III) and positive TB-PCR.histological findings compatible with TB (grade I, II, and III) with positive IGRA for MTB and followed by a successful response of anti-TB treatment [8].

### 2.3. Treatment of Tuberculous Lymphadenitis

The regimen of standard anti-TB treatment consisted of a combination of isoniazid (5 mg/kg/day or a total daily dose of 300 mg), rifampicin (10 mg/kg/day or a total daily dose of 600 mg), ethambutol (15 mg/kg/day), and pyrazinamide (25 mg/kg/day) once a day for the first 2 months. All drugs except pyrazinamide were administered for the following 4 months in accordance with the guidelines of the World Health Organization [9]. In the definitive regimen of anti-TB treatment, the same drug regimen was administered prior to the arrival of DST results from the laboratory, and replaced with the optimal drugs dictated by the DST results [17]. In both groups, anti-TB medication was not administered under clinical supervision, but rather was self-administered.

### 2.4. Definition of Recurrence after Treatment

The recurrence of TBL was classified into two subsets based on the method of diagnosis: microbiological recurrence or clinical recurrence. Microbiological recurrence was defined as re-detection of MTB after completion of treatment, as confirmed by culture results. Clinical recurrence was defined as lack of evidence of microbiological recurrence but regression of lymphadenopathy with anti-TB retreatment. Paradoxical response after treatment completion was defined as lack of evidence of microbiological recurrence and spontaneous regression of lymphadenopathy without anti-TB retreatment.

### 2.5. Statistical Analysis

The data are presented as the median and interquartile range (IQR) for continuous variables and numbers (percentages) for categorical variables. Data were compared using the Mann–Whitney *U* test for continuous variables and Pearson’s chi-squared test or Fisher’s exact test for categorical variables. The recurrence rate was analyzed using the Kaplan–Meier method. All tests were two-sided, and a *p*-value < 0.05 was considered significant. Data were analyzed using IBM SPSS Statistics, version 24 (IBM, Armonk, NY, USA).

## 3. Results

During the five-year study period, 281 patients were identified with TBL. Of these, 47 patients were excluded in accordance with the above criteria (Figure 1). A final total of 234 patients with TBL, consisting of 84 (35.9%) cases with definitive therapy based on DST profile and 150 (64.1%) cases with standard therapy were included and analyzed in this study. The demographic and clinical characteristics of enrolled cases are summarized in Table 1. There were 153 (65.4%) females with a median age of 46.0 (30.0–56.8) years. The number of enrolled patients with concomitant pulmonary TB was 36 (15.4%). The most common TBL site was the cervical region, followed by the mediastinal and axillary regions.

### 3.1. Clinical, Histological, and Microbiological Differences between the Definitive Therapy and Standard Therapy Groups

Table 2 shows the clinical, histological, and microbiological differences between the definitive therapy and empirical therapy groups. In terms of demographic, clinical, and histological characteristics, no significant differences were observed. Excisional biopsy followed by fine needle aspiration and core needle biopsy were used to clinch the diagnosis in 118 (50.4%) patients. The histologic findings were compatible with or highly suggestive of TBL in all cases, and there were no non-specific or inadequate specimens. In the microbiological analysis, the overall rate of positive culture using lymph node specimens was 61.3% (84/137). The frequency of AFB-smear positivity was greater in the definitive therapy group, suggesting a greater mycobacterial burden in that group. The median treatment duration for TBL was 8.6 months (6.3–9.7) and there was no difference between the two groups.

Among 84 cases with DST results (definitive therapy group), five (5.9%) had DR TBL; three were resistant to isoniazid only, one was resistant to isoniazid and protionamide, and the remaining one was resistant to isoniazid and streptomycin.

### 3.2. Comparison of Recurrence Rates after Treatment Completion between the Definitive Therapy and Standard Therapy Groups

A total of 234 patients with newly diagnosed TBL were followed for a median period of 28.0 months (24.0–43.0) after treatment completion. The duration of follow-up was not different in the two groups. During the follow-up period, nine cases (3.8%) had recurrence of TB after treatment completion. When they were identified as having recurrence, all of them were diagnosed with lymph node involvement, and with no other potential site of TB infection. They were all classified into the clinical recurrence subset; no case was placed in the microbiological recurrence subgroup (see classification criteria on *2.4. Definition of Recurrence after Treatment*). The median time to recurrence of TB after treatment completion was 21.6 months (12.2–37.9) (Table 3). Seven (4.7%) cases had recurrence of TB after treatment completion in the standard therapy group and two cases (2.4%) in the definitive therapy group, none of which had DR at the initial DST. The difference in recurrence rate between the two groups determined using the log–rank test was not statistically significant (Figure 2, *p* = 0.526). In addition, we tried to determine the risk factors of TBL recurrence, but this was not possible because of the small numbers involved.

In cases of a paradoxical response, 18 (7.7) experienced this phenomenon during therapy and three (1.3%) experienced it even after treatment completion. The time to paradoxical response was 2.0 months (1.6–3.3) during therapy and 3.3 months (3.3–15.4) after treatment completion. There were also no statistical differences between the two groups in the rate and occurrence time of paradoxical response.

Table 3 shows the clinical characteristics of the nine patients with TBL recurrence. They were treated again for 9.3 months (9.1–15.5) and followed for 29.5 (21.3–39.3) months. There were no further instances of recurrence or paradoxical response.

## 4. Discussion

We investigated the recurrence rate after treatment completion in TBL patients to evaluate the clinical outcome with or without DST results. The main findings of our study are that recurrence after TBL treatment occurred in 3.8% (9/234), and the rate was not statistically different in the two groups, the definitive therapy group versus the standard therapy group. In addition, during the follow-up period after treatment completion, there was no microbiological recurrence but only clinical recurrence in all nine such cases. This suggests that cases with culture-negative TBL can be treated with standard anti-TB drugs without any further invasive approach, although DST is not available but clinically stable after initiation of treatment. However, refractory drug-resistant TB infection such as rifampicin resistant, multi-DR or extensively DR TB should be looked upon separately and always managed by adequate definitive therapy.

According to 2005 to 2017 nationwide data from the Korea Centers for Disease Control and Prevention, 10.3 to 18.9% were diagnosed as EPTB. Among EPTB, TBL is the second most common cause, followed by tuberculous pleurisy [18]. Although the incidence of TBL has been decreasing slowly due to the decline in overall TB infection, approximately 1100 TBL cases are still detected annually in South Korea. However, EPTB presents a more challenging problem for the clinician than TBL due to the presence of several factors [19]:It has a low incidence, and therefore is less familiar to clinicians.The sites affected are less accessible.It occurs in vulnerable organs/tissues in spite of being characterized by a paucity of bacteria.

The combination of these characteristics poses a diagnostic and therapeutic problem.

In the diagnosis of TBL, developments in medical science and technology in the form of endobronchial ultrasound for mediastinal TBL, and endoscopy and laparoscopy for abdominal TBL permits efficient diagnosis [20,21]. Therefore, recent studies on TBL include more complex cases such as mediastinal and abdominal TBL [5,8,22]. In the present study, mediastinal and abdominal TBL were the 2nd and 4th most common causes, respectively. In addition, advancements in culture methods have also led to an improvement in diagnosis [23,24]. The therapeutic problem can also be overcome if we understand the characteristics of pauci-bacilli and apply the fundamental principles of anti-TB treatment. In this study, culture-negative TBL was treated in the same way as culture-negative PTB; we found that there was no microbiological recurrence but only clinical recurrence in all seven cases of recurrence (4.7%, 7/150) in the standard anti-TB therapy group, which was consistent with the results from previous studies on TBL [8,22].

When the culture result of a lymph node is finally negative after initiation of anti-TB treatment, it may indicate pauci-bacillary characteristics of TBL as EPTB rather than an inadequate study. However, clinicians conversely decided to extend the period of anti-TB treatment in many cases [1,3]. There are several reasons for this. Firstly, lymphadenopathy after anti-TB treatment, defined as a paradoxical response, is not uncommon in TBL. It usually occurs in the early period of anti-TB treatment, but can occur in the late period. Moreover, it can occur even after completion of anti-TB treatment [1,8,22]. Most cases of TBL involve the cervical lymph nodes so that clinicians can easily recognize deterioration of the condition with the unaided eye [1,5]. Such a worsening of the clinical status of the patient induces clinicians to administer their patients with more anti-TB treatment, and this explains the fact that several studies, including ours, report the continuation of therapy beyond the six-month recommendation, which is known to be equally effective [6,8,22,25]. However, the DR rate of EPTB is not different from that of PTB; therefore, culture-negative TBL can be treated with a standard short-course anti-TB regimen as used in culture-negative PTB. Secondly, there are no clear criteria that can gauge the response to anti-TB therapy in TBL. In PTB, baseline and follow-up evaluation using clinical, radiographic, and microbiologic methods is recommended to assess treatment response; these include evaluation of symptoms, periodic imaging, and monthly sputum smear and culture [9,26]. However, such precise guidelines for the assessment of response to treatment in TBL have not been formulated. As a result, clinical cues are the only option in the armory of the clinician faced with the task of quantifying the response to anti-TB therapy in TBL. Thirdly, no clear criteria have been put in place to aid clinicians in determining the best time for cessation of therapy.

In cases of TBL caused by non-DR mycobacteria, the recommendation is to use anti-TB drugs for 6 months as in PTB [9,17,26]. In addition, short duration of treatment with a standard drug regimen is also effective in culture-negative PTB and in TB pleurisy [27,28]. Therefore, culture-negative TBL can also be treated with standard therapy. In EPTB, some research has been conducted especially in tuberculous pleural effusion, strongly associated with PTB, the most common form and relatively easily accessible among EPTB [27,29,30]. Therefore, the treatment of TBL needs to be investigated further in order to lay down robust principles that conscientious clinicians can rely upon.

There were several limitations to this study. First, given its retrospective nature, selection bias may have influenced our findings. Secondly, it could not be determined whether cases of recurrence were a relapse or re-infection. However, the recurrence rate was not high, and not significantly different between the groups; further, this finding was similar to that of previous studies in South Korea [8,31]. Thus, we believe that it did not affect the findings of our study. Further prospective randomized controlled trials will therefore be necessary to discriminate between relapse and re-infection after treatment completion. Thirdly, there might be some uncertainty as to whether recurrence was a late paradoxical response rather than clinical recurrence, due to the absence of cases of microbiological recurrence. However, the definition of the recurrence and paradoxical response of TBL was identical to that used in previous studies so that the probability of misclassification was minimized. Fourthly, this study was conducted in an area of intermediate TB infection rate and very low HIV infection burden. There was only one HIV-infected case in the study screening, and this patient was excluded from our investigation, so this confounding factor was eliminated. Finally, anti-TB medication has not been practiced by directly observed treatment in South Korea. Therefore, our findings may not be generalized to other clinical situations, and further research is needed.

## 5. Conclusions

The current study provides new data on the treatment strategy of TBL. Culture-negative TBL can be appropriately treated with empirical standard treatment. The conclusions reached can provide helpful guidelines for the safe and efficacious treatment regimen in patients with culture-negative TBL, where doubts on the best management currently exist.

## Figures and Tables

**Figure 1 jcm-08-00813-f001:**
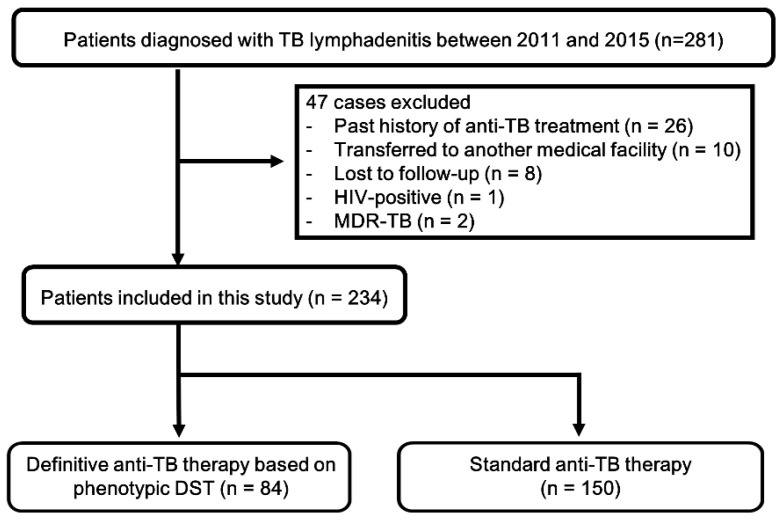
Flowchart of the study. TB, tuberculosis; HIV, human immune-deficiency virus; MDR, multi-drug resistant; DST, drug resistant test.

**Figure 2 jcm-08-00813-f002:**
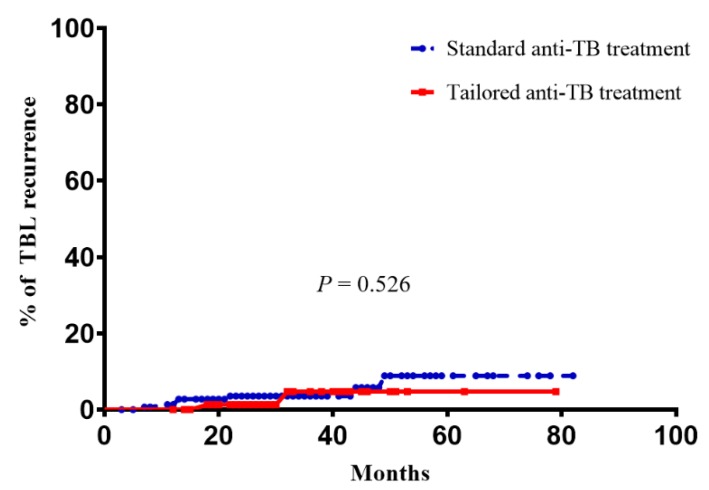
Comparison of recurrence rate between the definitive (*n* = 84) and standard (*n* = 150) anti-TB therapy groups in patients with TBL who completed anti-TB treatment. TBL, tuberculous lymphadenitis; TB, tuberculosis.

**Table 1 jcm-08-00813-t001:** Demographic and clinical characteristics of 234 patients with tuberculous lymphadenitis (TBL).

No. of Patients (%) or Median (IQR)	
Age, years, median (IQR)	46.0 (30.0–56.8)
Female	153 (65.4)
Comorbidity	
COPD and BA	4 (1.7)
Thyroid disease	7 (3.0)
Chronic liver disease	10 (4.3)
Diabetes	14 (6.0)
Cerebrovascular disease	7 (3.0)
Cardiovascular disease	5 (2.1)
Chronic kidney disease	8 (3.4)
Rheumatic disease	3 (1.3)
Malignancy	24 (10.3)
Hematologic disease	1 (0.4)
Neurologic disease	2 (0.9)
Combined TB in another site	
PTB	36 (15.4)
TB colitis	3 (1.3)
TB pleuritis	1 (0.4)
TB pericarditis	1 (0.4)
Location of affected lymph node	
Cervical	210 (89.7)
Mediastinal	11 (4.7)
Axillary	11 (4.7)
Abdominal	8 (3.4)
Submandibular	3 (1.3)
Periauricular	2 (0.9)
Inguinal	1 (0.4)

The data are presented as median (interquartile range) or No. (%). IQR, inter-quartile range; COPD, chronic obstructive lung disease; BA, bronchial asthma; TB, tuberculosis; PTB, pulmonary tuberculosis.

**Table 2 jcm-08-00813-t002:** Comparison of clinical feature and clinical outcomes between the definitive (*n* = 84) and standard (*n* = 150) anti-TB therapy groups.

	All Patients	Definitive	Standard	*p*-Value
	(*n* = 234)	(*n* = 84)	(*n* = 150)	
Age, years, median (IQR)	46.0 (30.0–56.8)	41.0 (30.0–54.3)	47.0 (31.8–58.0)	0.067
Female	153 (65.4)	50 (59.5)	103 (68.7)	0.158
Lymph node size	2.5 (2.0–3.6)	2.7 (2.1–3.8)	2.5 (2.0–3.6)	0.166
Diagnostic procedure				0.308
Excisional biopsy	118 (50.4)	37 (44.0)	81 (54.0)	
Core needle biopsy	44 (18.3)	19 (22.6)	25 (16.7)	
Fine needle aspiration biopsy	72 (30.8)	28 (33.3)	44(29.3)	
Histology				0.056
Gr I, epithelioid granuloma reaction with caseation	132 (56.4)	40 (47.6)	92 (61.3)	
Gr II, epithelioid granulomatous reaction without caseation	78 (33.3)	31 (36.9)	47 (31.3)	
Gr III, non-granulomatous reaction with necrosis	24 (10.3)	13 (15.5)	11 (7.3)	
Gr IV, non-specific				
Gr V, inadequate sample				
Microbiological examination				
MTB-PCR positive	202/218 (92.7)	75/84 (89.3)	127/134 (94.8)	0.003
AFB stain positive	36/180 (20.0)	23/80 (28.8)	13/100 (13.0)	<0.001
AFB culture positive	84/137 (61.3)	84/84 (100)	0/53 (0)	<0.001
IGRA positive	54/63 (85.7)	25/27 (92.6)	29/36 (80.6)	0.345
Treatment duration, months	8.6 (6.3–9.7)	7.7 (6.1–9.5)	8.8 (6.4–9.8)	0.207
After treatment completion				
Follow-up duration, months	28.0 (24.0–43.0)	27.0 (20.0–36.0)	29.0 (24.0–45.3)	0.079
Recurrence				
Microbiological recurrence	0	0	0	
Clinical recurrence	9 (3.8)	2 (2.4)	7 (4.7)	0.526
Paradoxical response				
Paradoxical response during treatment	18 (7.7)	3 (3.6)	15 (10.0)	0.122
Time to paradoxical response, months	2.0 (1.6–3.3)	1.7 (0.5–4.0)	2.0 (1.6–3.0)	0.441
Paradoxical response after treatment	3 (1.3)	2 (2.4)	1 (0.7)	0.263
Time to paradoxical response, months	3.3 (3.3–15.4)	3.3 (2.5–3.5)		0.157

The data are presented as median (interquartile range) or No. (%). IQR, inter-quartile range; Gr, grade; MTB, *Mycobacterium tuberculosis*; PCR, polymerase chain reaction; AFB, Acid-Fast Bacilli; IGRA, interferon-gamma release assay.

**Table 3 jcm-08-00813-t003:** Clinical characteristics and outcomes of the nine patients with TBL recurrence.

	All Patients
	(*n* = 9)
Time of recurrence after completion of treatment	21.6 (12.2–37.9)
Age	49.0 (24.0–54.5)
Female	7 (77.8)
Lymph node size at recurrence	2.3 (1.9–3.0)
Nature of recurrence of TBL	
New node	9 (100)
Enlargement of node at previous site(s)	1 (11.1)
New draining sinus	1 (11.1)
Histology	
Gr I, epithelioid granuloma reaction with caseation	8 (88.9)
Gr II, epithelioid granulomatous reaction without caseation	
Gr III, non-granulomatous reaction with necrosis	1 (11.1)
Gr IV, non-specific	
Gr V, inadequate sample	
Microbiological results of rebiopsy of affected lymph node	
MTB-PCR positive	9/9 (100)
AFB stain positive	0/9 (0)
AFB culture positive	0/9 (0)
Treatment duration, months	9.3 (9.1–15.5)
After treatment completion	
Follow-up duration, months	29.5 (21.3–39.3)
Recurrence	
Microbiological recurrence	0
Clinical recurrence	0

The data are presented as median (interquartile range) or No. (%). TBL, tuberculous lymphadenitis, Gr, grade; MTB, *Mycobacterium tuberculosis*; PCR, polymerase chain reaction; AFB, Acid-Fast Bacilli.

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
