# Peer review of "Clinical Characteristics and Treatment Outcomes of Definitive versus Standard Anti-Tuberculosis Therapy in Patients with Tuberculous Lymphadenitis"

_jcm, 2019, doi:10.3390/jcm8060813_

Reviewer 1 Report

Yousan Ko’s paper reported a comparison study of definitive therapy versus standard anti-tuberculosis therapy in patients with tuberculous lymphadenitis (TBL). There are limited data available regarding the treatment outcomes after standard anti-tuberculosis therapy in culture-negative TBL. This study provides new data on the treatment strategy of TBL, which is important to TBL treatment.

Their study was conducted in two groups, definitive (84 patients) and standard (150 patients) therapy. The recurrence rates are 2/84, 2.4% for definitive therapy and 7/150, 4.7% for standard therapy. The sample size here was small. But the recurrence rate was low and there is no big difference for two groups. The data they provided is well enough to reach their conclusion.

Overall, the work reported here is important, the paper is well written.

Author Response

Response to the Reviewers

We are submitting our point-by-point responses to the reviews and have revised our manuscript according to Reviwer`s comments. We have carefully considered each of the comments and criticisms offered by the reviewers. Specific changes are marked in highlight in this revised manuscript.

First Revision of jcm-525555

Reviewer #1

Point 1: Yousang Ko’s paper reported a comparison study of definitive therapy versus standard anti-tuberculosis therapy in patients with tuberculous lymphadenitis (TBL). There are limited data available regarding the treatment outcomes after standard anti-tuberculosis therapy in culture-negative TBL. This study provides new data on the treatment strategy of TBL, which is important to TBL treatment.

Their study was conducted in two groups, definitive (84 patients) and standard (150 patients) therapy. The recurrence rates are 2/84, 2.4% for definitive therapy and 7/150, 4.7% for standard therapy. The sample size here was small. But the recurrence rate was low and there is no big difference for two groups. The data they provided is well enough to reach their conclusion.

Overall, the work reported here is important, the paper is well written.

Response 1: We appreciate the reviewer’s comments

Reviewer 2 Report

This is a well written report of a multicentre retrospective study on lymphnode TB in South Korea.

The level of care is apparently beyond any doubt, and clinicians tended to treat longer than 6 months for these patients having a pauci-bacillary form of TB with hardly any drug resistance.

Follow-up seems to be very profound with only 8 patients lost to follow-up (flow chart, page 4).

The scientific debate is all about the definition of recurrence versus paradoxical inflammatory response. I agree that the authors used definitions derived from the literature, based on the time that new disease activity was noted; 47 were classified as paradoxical reactions and 9 as culture-negative relapse. I really wonder if  those 9 relapses were not just also paradoxical reactions. Again they use a formal definition to discriminate the two conditions -  paradoxical reactions, and culture-negative relapse; but perhaps the authors might try to argue in their Discussion that they have their doubts about what these 9 individuals had - relapse or paradoxical reactions.

So my only comment in about the Discussion Section; the data were well presented and the tables and pgraphs are very clear and helpful.

Below are the sections that might need some attention:

203               The main findings of our study are that recurrence after

204 TBL treatment occurred in 3.8% (9/234), and the rate was not statistically different in the two groups,

205 the definitive therapy group vs. the standard therapy group. In addition, during the follow-up period

206 after treatment completion, there was no microbiological recurrence but only clinical recurrence in

207 all 9 such cases.

The authors might add an alternative interpretation here - if the 9 were added to the 47paradoxical responses, these patients just received unnecessary repeat TB treatment; This aads to the overtreatment already discussed; i.e. 8 instead of 6 months that should actually suffice.

Below, the authors might be also more courageous; culture results being negetive after completion of treatment might in fact reflact exceelent treatment results in all patients. . .

234 When the culture result of a lymph node is finally negative after initiation of anti-TB treatment,      235 it may indicate pauci-bacillary characteristics of TBL as EPTB rather than an inadequate study. 236 However, clinicians conversely decided to extend the period of anti-TB treatment in many cases [1,3]. 237 There are several reasons for this. Firstly, lymphadenopathy after anti-TB treatment, defined as a 238 paradoxical response, is not uncommon in TBL. It usually occurs in the early period of anti-TB 239 treatment, but can occur in the late period. Moreover, it can occur even after completion of anti-TB 240 treatment [1,8,22].

The sentence below might need rephrasing: Korean doctors should perhaps need training in TBL; ofte paradoxical reactions seduce them to prolong or repeat TB treatrment, which is not necessary and also not beneficial;

Most cases of TBL involve the cervical lymph nodes so that clinicians can easily 241 recognize deterioration of the condition with the unaided eye [1,5].

Below:

 Such a worsening of the clinical 242 status of the patient is an indication for further anti-TB treatment, and this explains the fact that 243 several studies including ours report on the continuation of therapy beyond the 6-month 244 recommendation known to be equally effective [6,8,22,25]. NO THIS IS NOT AN INDICATION TO PROLONG TREATMENT!!

The following sentence:

262 retrospective data and the sample size was relatively not enough, despite data from multiple centers.

This statement is perhaps too modest; this is a large study with hardly any missing data so the retrospective design is not a big problem; there is no comparison between stopping treatment after 6 versus after 8 monhts, and no random to do watchful waiting versus trepeating TB treatment; that would help to resolve the scientific question whether these 9 'relapses' had relapse or paradoxical responses. 

164         ……………………..  Secondly, it could not be determined whether cases of

265 recurrence were a relapse or re-infection. However, the recurrence rate was not high, and not

266 significantly different between the groups; further, this finding was similar to that of previous studies 267 in South Korea [8,31].   GOOD POINT BU HERE PROPOSE AN RCT TO RESOLVE THE QUESTION

269         …………………………..  However, the time to post-TB

270 lymphadenopathy was much longer in the recurrence group (21.6 [12.2 – 37.9] months) than that in 271 the paradoxical response group (3.3 [3.3 – 15.4] months).  

THIS DOES NOT MAKE SENSE: THE DEFINITION OF THE TIME LINE WAS THE ONLY DIFFERENCE BETWEEN PARADOX AND RELAPSE SO THE IS A CIRCLE.. PLEASE CONSIDER REPHRASING

Typographical error:

Line 335 18. Annual Reoirt on the Notified Tuberculosis in . . Annual Report?

Author Response

Response to the Reviewers

We are submitting our point-by-point responses to the reviews and have revised our manuscript according to Reviwer`s comments. We have carefully considered each of the comments and criticisms offered by the reviewers. Specific changes are marked in highlight in this revised manuscript.

First Revision of jcm-525555

Reviewer #2

This is a well written report of a multicentre retrospective study on lymphnode TB in South Korea.

The level of care is apparently beyond any doubt, and clinicians tended to treat longer than 6 months for these patients having a pauci-bacillary form of TB with hardly any drug resistance.

Follow-up seems to be very profound with only 8 patients lost to follow-up (flow chart, page 4).

The scientific debate is all about the definition of recurrence versus paradoxical inflammatory response. I agree that the authors used definitions derived from the literature, based on the time that new disease activity was noted; 47 were classified as paradoxical reactions and 9 as culture-negative relapse. I really wonder if  those 9 relapses were not just also paradoxical reactions. Again they use a formal definition to discriminate the two conditions -  paradoxical reactions, and culture-negative relapse; but perhaps the authors might try to argue in their Discussion that they have their doubts about what these 9 individuals had - relapse or paradoxical reactions.

So my only comment in about the Discussion Section; the data were well presented and the tables and graphs are very clear and helpful.

Point 1: Below are the sections that might need some attention:

(Line 203-207) The main findings of our study are that recurrence after TBL treatment occurred in 3.8% (9/234), and the rate was not statistically different in the two groups, the definitive therapy group vs. the standard therapy group. In addition, during the follow-up period after treatment completion, there was no microbiological recurrence but only clinical recurrence in all 9 such cases.

The authors might add an alternative interpretation here - if the 9 were added to the 47 paradoxical responses, these patients just received unnecessary repeat TB treatment; This adds to the overtreatment already discussed; i.e. 8 instead of 6 months that should actually suffice.

Response 1: We appreciate the reviewer’s comments. We understand the reviewer’s concern. But I think there must be some misunderstanding. I wound appreciated it if you could check the bottom of Table 2 once more. The number of paradoxical reactions is 21 cases (18 cases during treatment and only 3 cases after treatment, bottom of Table 2, page 6). That 47 cases are the number of excluded cases because of several reasons of exclusion (flow chart, page 4). The paradoxical response, especially in cases after anti-TB treatment, might confuse with the recurrence of TBL. But, the number of paradoxical responses after treatment completion is only 3 in our study. It is relatively few compared with 9 case as the number of recurrences. Thus, we thank that the paradoxical response after treatment completion did not affect the findings of our study as low recurrence rate.

Point 2:. Below, the authors might be also more courageous; culture results being negative after completion of treatment might in fact reflect excellent treatment results in all patients. . .

(Line 234-240) When the culture result of a lymph node is finally negative after initiation of anti-TB treatment, it may indicate pauci-bacillary characteristics of TBL as EPTB rather than an inadequate study. However, clinicians conversely decided to extend the period of anti-TB treatment in many cases [1,3]. There are several reasons for this. Firstly, lymphadenopathy after anti-TB treatment, defined as a paradoxical response, is not uncommon in TBL. It usually occurs in the early period of anti-TB treatment, but can occur in the late period. Moreover, it can occur even after completion of anti-TB treatment [1,8,22].

The sentence below might need rephrasing: Korean doctors should perhaps need training in TBL; often paradoxical reactions seduce them to prolong or repeat TB treatrment, which is not necessary and also not beneficial;

(Line 240-241) Most cases of TBL involve the cervical lymph nodes so that clinicians can easily recognize deterioration of the condition with the unaided eye [1,5].

Below:

(Line 241-244) Such a worsening of the clinical status of the patient is an indication for further anti-TB treatment, and this explains the fact that several studies including ours report on the continuation of therapy beyond the 6-month recommendation known to be equally effective [6,8,22,25]. NO THIS IS NOT AN INDICATION TO PROLONG TREATMENT!!

Response 2: We appreciate the reviewer’s comments. We understand the reviewer’s concern and absolutely agree. We revised the manuscript according to your comments as below.

Such a worsening of the clinical status of the patient makes clinicians to decide his patients more anti-TB treatment.

Point 3: The following sentence:

(Line 262) retrospective data and the sample size was relatively not enough, despite data from multiple centers.

This statement is perhaps too modest; this is a large study with hardly any missing data so the retrospective design is not a big problem; there is no comparison between stopping treatment after 6 versus after 8 monhts, and no random to do watchful waiting versus trepeating TB treatment; that would help to resolve the scientific question whether these 9 'relapses' had relapse or paradoxical responses. 

Response 3: We appreciate the reviewer’s comments. We revised the manuscript according to your comments as below.

First, given its retrospective nature, selection bias may have influenced our findings.

Point 4:.

Line 264

Secondly, it could not be determined whether cases of recurrence were a relapse or re-infection. However, the recurrence rate was not high, and not significantly different between the groups; further, this finding was similar to that of previous studies in South Korea [8,31].   GOOD POINT BU HERE PROPOSE AN RCT TO RESOLVE THE QUESTION

Response 4: We appreciate the reviewer’s comments and absolutely agree. We revised the manuscript according to your comments as below.

Further prospective randomized controlled trials will therefore be necessary to discriminate between relapse and re-infection after treatment completion.

Point 5:.

(Line 269-  However, the time to post-TB lymphadenopathy was much longer in the recurrence group (21.6 [12.2 – 37.9] months) than that in the paradoxical response group (3.3 [3.3 – 15.4] months). THIS DOES NOT MAKE SENSE: THE DEFINITION OF THE TIME LINE WAS THE ONLY DIFFERENCE BETWEEN PARADOX AND RELAPSE SO THE IS A CIRCLE.. PLEASE CONSIDER REPHRASING

Response 6: We understand the reviewer’s concern and absolutely agree. We revised the manuscript according to your comments as below

Thirdly, there might be some uncertainty as to whether recurrence was a late paradoxical response rather than clinical recurrence, due to the absence of cases of microbiological recurrence. However, the definition of the recurrence and paradoxical response of TBL was identical to that used in previous studies so that the probability of misclassification was minimized.

Point 6: Typographical error:

Line 335 18. Annual Reoirt on the Notified Tuberculosis in . . Annual Report?

Response 6: We apologize for our careless. We corrected this reference as a highlight.